# Congenital transmission of Chagas disease: The role of newborn therapy on the disease's dynamics

**Meriem Boukaabar[1], Bismark Oduro[1], Paul Chataa[2]***

**1** Department of Mathematics, Pennsylvania Western University, California, PA, United States of America,
**2** Department of Mathematics, University of Cape Coast, Cape Coast, Ghana

\* paul.chataa@stu.ucc.edu.gh

**Data Availability Statement:** The research was not based on data. We estimated our parameter values from literature that was duly cited in the manuscript. Therefore, all relevant data are included in the manuscript.

## Abstract

Chagas disease, also known as American trypanosomiasis, is caused by a protozoan blood-borne pathogen called Trypanosoma cruzi. The World Health Organization (WHO) has classified Chagas as one of 21 neglected tropical diseases present in the world and estimates that 6-7 million people are currently infected with Chagas. Congenital transmission of Chagas disease contributes to a significant amount of new infections, especially in endemic areas where 22.5% of new infections are due to congenital transmission. In this paper, we investigate congenital transmission's impact on Chagas disease dynamics through a mathematical model. Specifically, we examine how treating a proportion of infants born to infected individuals impacts the progression and spread of Chagas disease. The influence of newborn therapy on the dynamics of the model is thoroughly investigated, both theoretically and numerically. The results illustrate the importance of treating a high proportion of newborns to reduce the number of infected cases of the disease. The findings show that the therapy given to newborns is necessary but not sufficient to curb the transmission of Chagas disease, and a comprehensive approach that includes vector and vertical transmission control strategies is essential for eradicating Chagas disease. We also observed that if vector transmission can be controlled, then at least 55% of the newborns need to be treated to eliminate the disease.

## Introduction

Chagas disease, also known as American trypanosomiasis, is an anthropozoonosis disease caused by a protozoan blood-borne pathogen called Trypanosoma cruzi [1]. The disease is predominantly active in Latin America, where it is a major public health issue [2]. The World Health Organization (WHO) has classified Chagas as one of 21 neglected tropical diseases in the world [3]. Additionally, the WHO estimates that 6-7 million people are currently infected with Chagas, and 75 million people are at risk for acquiring the disease [4]. Higher incidence rates are typically associated with areas that have poorly constructed housing, which serve as hiding places for the insect vectors that transmit the disease [2]. The

**Funding:** The author(s) received no specific funding for this work.

**Competing interests:** The authors have declared that no competing interests exist.

disease is vectorized by Triatomine (reduviid) bugs, also known as "kissing bugs," because they bite the host around their lips when they feed [5]. When the bug feeds on humans, it defecates, which allows the T. cruzi to exit with the feces and enter the host's body [6]. This is one of the most common routes of infection. Other common routes of infection include congenital transmission, consumption of triatomine insects, needle sharing, and transfusional transmission [7]. The highest number of new acute infection cases comes from vector and congenital transmission [8]. This is especially true in Latin American countries, where approximately 22.5% of new infections are due to congenital transmission [9]. In both the acute and chronic phases of the infection, the disease can be transmitted from the infected mother through the placenta to the embryo or fetus [10]. In 1–10% of infants of infected mothers, congenital T. cruzi infection occurs [2]. Pregnant-infected women typically have higher rates of premature births and miscarriages [11]. Typically, the mothers and the infected children are asymptomatic, which makes the diagnosis of Chagas challenging [12]. Even in cases where symptoms are present, they are non-specific symptoms like fevers, swollen lymph nodes, and hepatosplenomegaly [13]. Regardless of the symptoms, all untreated infected infants are at a 20–30% risk of developing severe cardiac and intestinal complications later on in their life [14]. Ultimately, the faster the diagnosis and subsequent treatment, the more effective it is [10]. There is a 90–95% cure rate when the disease is recognized early, and treatment is used [15]. As infected patients age, the cure rate decreases, so diagnosis should be a priority [16]. Current diagnosis and treatment methods consist of a multistep method. Firstly, detection requires maternal serological screening [17]. Typically, if a mother tests positive two times in a row, the newborns are suspected of having Chagas disease and are tested [2]. The most common method for testing newborns is examining cord blood from their seropositive mothers using microscopy (also called the "micro method") or polymerase chain reaction (PCR) techniques, anytime until they are one-month old [18]. Infants who test positive via microscopy or PCR are considered to have Chagas disease [17]. Unfortunately, these methods are unreliable; more than 50% of infections are not recognized by microscopy [19]. Therefore, infants who are tested after one month of birth or test negative are retested using serology when they are 9-12 months old [2]. Treatment options include Benznidazole or Nifurtimox [4]. Treatment during pregnancy is not currently recommended because of a lack of data on safety [16]. However, since the treatment options are highly effective for newborns, treatment should begin as soon as the diagnosis is made [2].

A few articles have been published on using mathematical models to explore Chagas disease dynamics, including [20–23]. Raimundo et al. in [24] focused on the congenital transmission of Chagas disease in populations where vectorial transmission has been eliminated. By considering both vertical transmission and the presence of vectorial transmission, our study expanded on this work. Furthermore, we also incorporated vector control measures into the model. Coffield et al. in [25] used a mathematical model to explore Chagas disease transmission by including congenital and oral transmission modes in humans and domestic mammals. The research concluded that while congenital transmission has a limited impact on infection, oral transmission in domestic mammals significantly contributes to the disease's spread, highlighting the importance of considering alternative transmission modes in disease control strategies [25]. None of these published papers explored the impact of congenital transmission and newborn therapy in controlling the disease. Our study addresses this gap by investigating the impact of congenital transmission and newborn therapy on controlling Chagas disease. The main question guiding our research is: how do congenital transmission and newborn therapy influence Chagas disease's dynamics, particularly in controlling its spread?

## Materials and methods

In this section, we develop a compartmental model that reflects the dynamics of Chagas disease in both human and vector populations. The vector population is divided into two classes at time $t$: susceptible vectors ($S_v$) and infected vectors ($I_v$). The human population is divided into four classes at time $t$: infected acute humans ($I_a$), infected chronic humans ($I_c$), susceptible humans ($S_h$), and newborn babies from infected mothers ($M$). The natural death rate of vectors is denoted by $\mu_v$, and the model assumes all newborns from non-infected mothers are susceptible. If a susceptible vector feeds on an infected acute or infected chronic human, the rate of disease transmission from the human to the vector is represented by $\beta_{hv}$, and the susceptible vector moves to the infected vector class and stays there for life. When an infected vector bites a susceptible human, the disease is transmitted at a rate of $\beta_{vh}$, and the susceptible human is moved to the infected acute stage. From there, the infected acute human can progress to the infected chronic human class; this progression rate is denoted by $k$. Individuals in the infected chronic class remain in that class for life unless they leave the population through natural death at rate $\mu_h$ or the death rate from Chagas given by $\delta_h$. The birth rate of infected acute and infected chronic mothers combined, represented by $b_h(I_c + I_a)$, is what comprises the $M$ class. Newborns to infected mothers in the $M$ class progress to either the $S_h$ or $I_a$ class at rate $p$; the progression could be due to treatment or aging. In [15], the authors demonstrated that the treatment of newborns is effective, with a 90–95% efficacy. We incorporate newborn therapy and let $0 \leq \alpha \leq 1$ represent the proportion of newborns (to infected mothers) who receive successful therapy and move to the class $S_h$. The remaining proportion, $1 - \alpha$, undergo unsuccessful treatment or do not receive any treatment and are classified as infected acute individuals. Values of $\alpha$ close to unity imply that almost all newborns receive or undergo perfect therapy, while a low level of $\alpha$ implies that no newborn receives treatment therapy. A diagram depicting the dynamics explained in this section is shown in Fig 1; the parameters and their meanings are presented in Table 1.

Under the assumptions described above and the diagram, we obtain the following system of nonlinear ordinary differential equations.

$$
\begin{aligned}
\frac{dS_v}{dt} &= b_v - \lambda_v S_v - \mu_v S_v \\
\frac{dI_v}{dt} &= \lambda_v S_v - \mu_v I_v \\
\frac{dS_h}{dt} &= b_h + \alpha p M - \lambda_h S_h - \mu_h S_h \\
\frac{dI_a}{dt} &= \lambda_h S_h + (1 - \alpha) p M - (k + \mu_h) I_a \\
\frac{dI_c}{dt} &= k I_a - (\mu_h + \delta_h) I_c \\
\frac{dM}{dt} &= b_h(I_a + I_c) - (p + \mu_h) M
\end{aligned}
\tag{1}
$$

Where

$$
\lambda_v = \beta_{hv}\left(\frac{I_a + I_c}{N_h}\right), \quad \lambda_h = \left(\frac{\beta_{vh} I_v}{N_v}\right), \quad N_v = S_v + I_v, \quad N_h = S_h + I_a + I_c + M.
$$

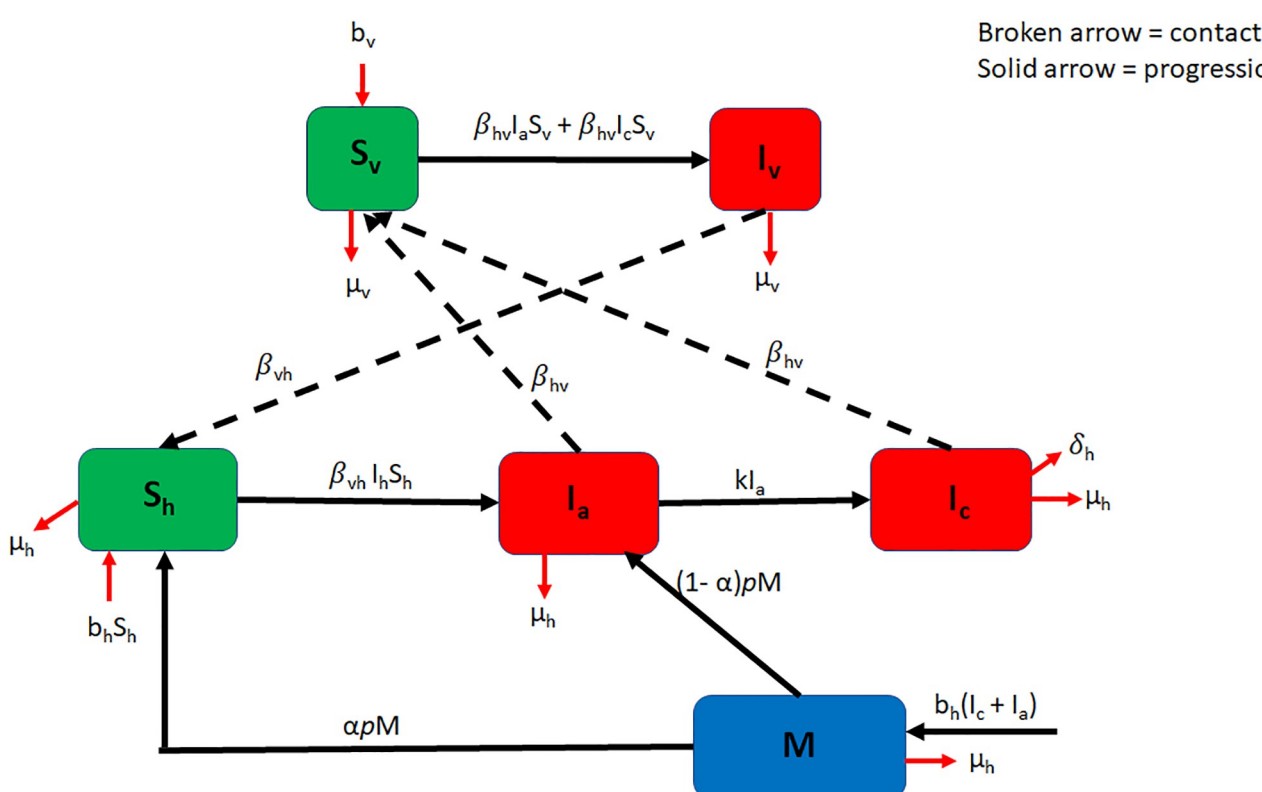

**Fig 1. Schematic diagram of the model.** See Table 1 for meanings of parameters/variables.

The disease-free equilibrium (DFE) is given as:

$$DFE = \left( \frac{b_v}{\mu_v}, 0, \frac{b_h}{\mu_h}, 0, 0, 0 \right). \tag{2}$$

A next-generation approach is defined as the dominant eigenvalue (spectral radius) of the matrix $FV^{-1}$ [26–28], where $F$ and $V^{-1}$ are matrices determined as: $F = \left[ \frac{\partial F_i(x_0)}{\partial x_j} \right]$ and

**Table 1. Parameters of the disease model and their meanings.**

| Parameter | Meaning |
| --- | --- |
| $\beta_{vh}$ | The transmission rate from infected vectors to a susceptible human |
| $\beta_{hv}$ | The transmission rate from infected human to susceptible vectors |
| $\mu_h$ | Death from unrelated causes rate of human |
| $\mu_v$ | Death rate of vectors |
| $\delta_h$ | Death rate from the disease of humans in the chronic stage |
| $b_h$ | Birth rate of humans |
| $b_v$ | Recruitment rate of the vectors. This depends on an available blood meal, including birds and alternative hosts |
| $k$ | The progression rate from infected human in the acute to the chronic stage |
| $p$ | Progression rate from $M$ to $S_h$ or $I_a$ class |
| $\alpha$ | Proportion of newborns that undergo treatment therapy |

$V = \left[\frac{\partial V_i(x_0)}{\partial x_j}\right]$. Here, $x_j$ is the number of infested units, $x_0$ is the disease-free equilibrium, $F_i$ is the rate of appearance of new infection in the infected compartments, $V_i = V_i^- - V_i^+$ with $V_i^-$ denoting the rate at which infected individuals are transferred out of the infected compartments and $V_i^+$ denoting the rate at which individuals are transferred into the infected compartments.

We will use the next-generation method to compute the control reproduction number $\mathcal{R}_c$.

$$F_i = \begin{bmatrix} F_{I_v} \\ F_{I_a} \\ F_{I_c} \\ F_M \end{bmatrix} = \begin{bmatrix} \lambda_v S_v \\ \lambda_h S_h \\ 0 \\ b_h(I_a + I_c) \end{bmatrix}$$

and

$$V_i = \begin{bmatrix} V_{I_v} \\ V_{I_a} \\ V_{I_c} \\ V_M \end{bmatrix} = \begin{bmatrix} \mu_v I_v \\ -(1-\alpha)pM + (k+\mu_h)I_a \\ -kI_a + (\mu_h + \delta_h)I_c \\ (p+\mu_h)M \end{bmatrix}.$$

Therefore

$$F = \begin{bmatrix} 0 & \frac{\beta_{hv}S_v^*}{N_h^*} & \frac{\beta_{hv}S_v^*}{N_h^*} & 0 \\ \frac{\beta_{vh}S_h^*}{N_v^*} & 0 & 0 & 0 \\ 0 & 0 & 0 & 0 \\ 0 & b_h & b_h & 0 \end{bmatrix}.$$

and

$$V = \begin{bmatrix} \mu_v & 0 & 0 & 0 \\ 0 & k+\mu_h & 0 & -(1-\alpha)p \\ 0 & -k & \mu_h + \delta_h & 0 \\ 0 & 0 & 0 & p+\mu_h \end{bmatrix},$$

where $S_h^* = \frac{b_h}{\mu_h} = N_h^*$ and $S_v^* = \frac{b_v}{\mu_v} = N_v^*$.

By the next generation method, the control reproduction number is the spectral radius $FV^{-1}$. That is,

$$\mathcal{R}_c = \frac{\Psi_1 + \sqrt{(k+\mu_h+\delta_h)\Psi_2 + \Psi_1^2}}{2(\mu_h+k)(\mu_h+p)(\mu_h+\delta_h)},$$

where

$$\begin{aligned} \Psi_1 &= b_h p(1-\alpha)(k+\mu_h+\delta_h) \\ \Psi_2 &= 4\beta_{hv}\beta_{vh}(\mu_h+k)(\mu_h+p)^2(\mu_h+\delta_h)/\mu_v. \end{aligned}$$

Let us consider a scenario that allows a perfect treatment that is $\alpha = 1$.

$$\mathcal{R}_L = \left( \lim_{\alpha \to 1} \mathcal{R}_c \right)^2 = \frac{\beta_{hv}\beta_{vh}(k + \mu_h + \delta_h)}{\mu_v(k + \mu_h)(\mu_h + \delta_h)}. \tag{3}$$

Eq 3 shows that perfect treatment of newborns is insufficient to exterminate the infection if $\mathcal{R}_L > 1$. In addition to the newborn therapy, control measures that would reduce $\mathcal{R}_L$ below unity are required to eradicate the disease.

Let us consider the existence of an endemic equilibrium of the system. The endemic equilibrium levels are the steady-state solutions where the Chagas infection persists in the population. So, at the Chagas disease persistence equilibrium, $EE = (S_v^*, I_v^*, S_h^*, I_a^*, I_c^*, M^*)$, the following equations are satisfied:

$$
\begin{aligned}
0 &= b_v - \lambda_v^* S_v^* - \mu_v S_v^*, \\
0 &= \lambda_v^* S_v^* - \mu_v I_v^*, \\
0 &= b_h + \alpha p M^* - \lambda_h^* S_h^* - \mu_h S_h^*, \\
0 &= \lambda_h S_h^* + (1 - \alpha) p M^* - (k + \mu_h) I_a^*, \\
0 &= k I_a^* - (\mu_h + \delta_h) I_c^*, \\
0 &= b_h (I_a^* + I_c^*) - (p + \mu_h) M^*.
\end{aligned}
\tag{4}
$$

Also, at the endemic equilibrium, the force of infections are $\lambda_v^* = \beta_{hv} \left( \frac{I_a^* + I_c^*}{N_h^*} \right) \leq \beta_{hv}$, $\lambda_h^* = \left( \frac{\beta_{vh} I_v^*}{N_v^*} \right) \leq \beta_{vh}$. To explore the effects of $\alpha$ on the infected individuals at equilibrium theoretically, we consider when $\lambda_v^* = \beta_{hv}$ and $\lambda_h^* = \beta_{vh}$ to obtain explicit equations for the endemic levels of the state variables. By the first, second, third, fifth, and sixth lines of (4), we have

$$
\begin{aligned}
S_v^* &= \frac{b_v}{\beta_{hv} + \mu_v}, \\
I_v^* &= \frac{\beta_{hv} b_v}{\mu_v(\beta_{hv} + \mu_v)}, \\
I_c^\star &= \frac{k I_a^\star}{\mu_h + \delta_h}, \\
M^\star &= \frac{b_h(I_a^* + I_c^*)}{p + \mu_h} = \frac{b_h(\mu_h + \delta_h + k) I_a^\star}{(p + \mu_h)(\mu_h + \delta_h)}, \\
S_h^* &= \frac{b_h}{\beta_{vh} + \mu_h} + \frac{\alpha p M_a^\star}{\beta_{vh} + \mu_h} = \frac{b_h}{\beta_{vh} + \mu_h} + \frac{\alpha p b_h(\mu_h + \delta_h + k) I_a^\star}{(\mu_h + \delta_h)(\mu_h + p)(\beta_{vh} + \mu_h)}.
\end{aligned}
$$

Also, by the fourth line of (4), we have

$$I_a^* = \frac{\beta_{vh} S_h^* + (1 - \alpha) p M^*}{k + \mu_h}. \tag{5}$$

Substituting the expressions of $S_h^*$ and $M^*$ into (5) and solving for $I_a^*$ gives

$$I_a^\star = \frac{b_h \beta_{vh}}{(k + \mu_h)(\beta_{vh} + \mu_h)(1 - \Phi)},$$

where

$$\Phi = \frac{b_h p (\mu_h + \delta_h + k)(\beta_{vh} + (1 - \alpha)\mu_h)}{(k + \mu_h)(\beta_{vh} + \mu_h)(\mu_h + \delta_h)(p + \mu_h)}.$$

Hence, an endemic equilibrium exists when $\Phi < 1$. The parameter $\alpha$ is our control parameter; let us describe how it influences the endemic equilibrium. The $\frac{\partial I_a^*}{\partial \alpha} < 0$; indicating that $I^*$ is always decreasing with respect to $\alpha$. Observe that $I_c^*$ and $M^*$ also decrease with respect to $\alpha$, as they are directly proportional to $I_a^*$. This implies that treating many infants can significantly lower the endemic prevalence.

## Stability analysis of the disease-free equilibrium

In this section, we determine the local and global stability of the disease-free equilibrium.

First, we determine the local stability of the disease-free equilibrium by computing the eigenvalues of the linearized Jacobian matrix at the disease-free equilibrium and obtain

$$J_0(DFE) = \begin{pmatrix} -\mu_v & 0 & 0 & -\frac{\beta_{hv}S_v^\star}{N_h^\star} & -\frac{\beta_{hv}S_v^\star}{N_h^\star} & 0 \\ 0 & -\mu_v & 0 & \frac{\beta_{hv}S_v^\star}{N_h^\star} & \frac{\beta_{hv}S_v^\star}{N_h^\star} & 0 \\ 0 & -\frac{\beta_{vh}S_h^\star}{N_v^\star} & -\mu_h & 0 & 0 & \alpha p \\ 0 & \frac{\beta_{vh}S_h^\star}{N_v^\star} & 0 & -(k + \mu_h) & 0 & (1 - \alpha)p \\ 0 & 0 & 0 & k & -(\mu_h + \delta_h) & 0 \\ 0 & 0 & 0 & b_h & b_h & -(\mu_h + p) \end{pmatrix}.$$

The eigenvalues of the Jacobian matrix, $J_0(DFE)$ are, $\varrho_1 = -\mu_v$, $\varrho_2 = -\mu_h$, $\varrho_3 = -\mu_v$, $\varrho_4 = -(\mu_h + p)$, $\varrho_5 = -(\mu_h + \delta_h)$, and

$$\varrho_6 = \frac{\mu_v(p + \mu_h)(k + \mu_h)(\mu_h + \delta_h)}{(p + \mu_h)B_1 B_2 + \mu_v b_h (1 - \alpha)p}(\varepsilon - 1)$$

where

$$\varepsilon = \frac{((p + \mu_h)B_1 B_2 + \mu_v b_h (1 - \alpha)p)(\mu_h + \delta_h + k)}{\mu_v(p + \mu_h)(k + \mu_h)(\mu_h + \delta_h)}.$$

The disease-free state of the system is locally asymptotically stable when $\varepsilon < 1$.

Observe that for $\alpha = 1$, we have

$$\varrho_6 = \frac{\mu_v(k + \mu_h)(\mu_h + \delta_h)}{B_1 B_2}(\mathcal{R}_L - 1),$$

leading to the following result.

**Theorem 1**. *For $\alpha = 1$, the disease-free equilibrium of the system is locally asymptotically stable if $\mathcal{R}_L < 1$ and unstable if $\mathcal{R}_L > 1$. For $0 \leq \alpha < 1$, the disease-free equilibrium of the system is locally asymptotically stable if $\varepsilon < 1$ and unstable if $\varepsilon > 1$.*

Next, we will apply the approach of Castillo-Chavez et al [29] to prove the global stability of the disease-free equilibrium. The approach is defined in the theorem below.

**Theorem 2**. *If a model system can be written in the form*:

$$\frac{dX}{dt} = F(X, 0), \quad \frac{dI}{dt} = G(X, I), \quad G(X, 0) = 0,$$

where $X \in \Re^m$ denotes the number of uninfected compartments and $I \in \Re^n$, denotes the number of infected compartments, including latent, exposed, and acute individuals. $U(X^\star, 0)$ denotes the disease-free equilibrium of the system. Then the conditions $(\mathfrak{H}_1)$ and $(\mathfrak{H}_2)$ must be satisfied to guarantee local asymptotic stability.

$\mathfrak{H}_1$: For $\frac{dX}{dt} = F(X, 0)$, $X^\star$ is globally asymptotically stable.

$\mathfrak{H}_2$: $G(X, I) = AI - \hat{G}(X, 0) \geq 0$ for $(X, I) \in \Delta$, where $A = D_I G(X^\star, 0)$ is a Metzler matrix (the off-diagonal elements of $A$ are non-negative) and $\Delta$ is the region where the model makes biological sense and mathematically well-posed. Then the fixed point $U_0 = (X^\star, 0)$ is globally asymptotically stable equilibrium of the Chagas infection model provided $\mathcal{R}_c < 1$.

**Theorem 3**. *The disease-free equilibrium*

$$DFE = \left( \frac{b_v}{\mu_v}, 0, \frac{b_h}{\mu_h}, 0, 0, 0 \right)$$

*is globally asymptotically stable if the conditions $(\mathfrak{H}_1)$ and $(\mathfrak{H}_2)$ are satisfied.*

*Proof.* From the model system, we have $X \in \Re^2 = (S_v^\star, S_h^\star)$ and $I \in \Re^4 = (I_v^\star, I_a^\star, I_c^\star, M^\star)$. Hence, for condition $(\mathfrak{H}_1)$, we have

$$\frac{dX}{dt} = F(X, 0) = \begin{pmatrix} b_v - \frac{\beta_{hv} S_v I_a}{N_v} - \frac{\beta_{hv} S_v I_c}{N_v} - \mu_v S_v \\ b_h + \alpha p M - \frac{\beta_{vh} S_h I_v}{N_h} - \mu_h S_h \end{pmatrix}$$

and

$$\frac{dI}{dt} = G(X, I) = \begin{pmatrix} \frac{\beta_{hv} S_v I_a}{N_v} + \frac{\beta_{hv} S_v I_c}{N_v} - \mu_v I_v \\ \frac{\beta_{vh} S_h I_v}{N_h} + (1 - \alpha) p M - (k + \mu_h) I_a \\ k I_a - (\mu_h + \delta_h) I_c \\ b_h I_a + b_h I_c - (p + \mu_h) M \end{pmatrix}.$$

It follows that

$$F(X, 0) = \begin{pmatrix} -\mu_v & 0 \\ 0 & -\mu_h \end{pmatrix}.$$

The eigenvaules from the matrix $F(X, 0)$ are obtained to be

$$\pi_1 = -\mu_v < 0,$$
$$\pi_2 = -\mu_h < 0.$$

Since all the eigenvalues of the matrix $F(X, 0)$ are negative, it follows that $X^\star$ is always globally asymptotically stable. Also, applying Theorem (2) to the Chagas disease model system

gives

$$\hat{G}(X,I) = AI - G(X,I)$$

$$= \begin{pmatrix} 0 & \frac{\beta_{hv}S_v^\star}{N_v^\star} & \frac{\beta_{hv}S_v^\star}{N_v^\star} & 0 \\ \frac{\beta_{vh}S_h^\star}{N_h^\star} & -(k+\mu_h) & 0 & -(1-\alpha)p \\ 0 & k & -(\mu_h+\delta_h) & 0 \\ 0 & b_h & b_h & -(p+\mu_h) \end{pmatrix} \begin{pmatrix} I_v \\ I_a \\ I_c \\ M \end{pmatrix}$$

$$- \begin{pmatrix} \frac{\beta_{hv}S_vI_a}{N_v} + \frac{\beta_{hv}S_vI_c}{N_v} - \mu_vI_v \\ \frac{\beta_{vh}S_hI_v}{N_h} + (1-\alpha)pM - (k+\mu_h)I_a \\ kI_a - (\mu_h+\delta_h)I_c \\ b_hI_a + b_hI_c - (p+\mu_h)M \end{pmatrix}$$

Hence,

$$\hat{G}(X,I) = \begin{pmatrix} \frac{\beta_{hv}S_v^\star I_a}{N_v^\star} + \frac{\beta_{hv}S_v^\star I_c}{N_v^\star} \\ \frac{\beta_{vh}S_h^\star I_v}{N_h^\star} - (k+\mu_h)I_a - (1-\alpha)pM \\ kI_a - (\mu_h+\delta_h)I_c \\ b_hI_a + b_hI_c - (p+\mu_h)M \end{pmatrix}$$

$$- \begin{pmatrix} \frac{\beta_{hv}S_vI_a}{N_v} + \frac{\beta_{hv}S_vI_c}{N_v} - \mu_vI_v \\ \frac{\beta_{vh}S_hI_v}{N_h} + (1-\alpha)pM - (k+\mu_h)I_a \\ kI_a - (\mu_h+\delta_h)I_c \\ b_hI_a + b_hI_c - (p+\mu_h)M \end{pmatrix}$$

Therefore,

$$\hat{G}_0(X,I) = \begin{pmatrix} \left[\frac{\beta_{hv}S_v^\star I_a}{N_v^\star} - \frac{\beta_{hv}S_vI_a}{N_v}\right] + \left[\frac{\beta_{hv}S_v^\star I_c}{N_v^\star} - \frac{\beta_{hv}S_vI_c}{N_v}\right] + \mu_vI_v \\ \frac{\beta_{vh}S_h^\star I_v}{N_h^\star} - \frac{\beta_{vh}S_hI_v}{N_h} \\ 0 \\ 0 \end{pmatrix}$$

$$= \begin{pmatrix} \frac{\beta_{hv}I_a}{N_v^\star}\left[S_v^\star - S_v\right] + \frac{\beta_{hv}I_c}{N_v^\star}\left[S_v^\star - S_v\right] \\ \frac{\beta_{vh}I_v}{N_h^\star}\left[S_h^\star - S_h\right] \\ 0 \\ 0 \end{pmatrix}.$$

So, $A$ is a Metzler matrix with non-negative off-diagonal elements. We observed that

$$\hat{G}_0(X, I) = \begin{pmatrix} \frac{\beta_{hv}I_a}{N_v^\star}\left[S_v^\star - S_v\right] + \frac{\beta_{hv}I_c}{N_v^\star}\left[S_v^\star - S_v\right] \\ \frac{\beta_{vh}I_v}{N_h^\star}\left[S_h^\star - S_h\right] \\ 0 \\ \\ 0 \end{pmatrix} \geq 0,$$

because $\frac{\beta_{hv}I_a}{N_v^\star}\left[S_v^\star - S_v\right] + \frac{\beta_{hv}I_c}{N_v^\star}\left[S_v^\star - S_v\right] \geq 0$ and $\frac{\beta_{vh}I_v}{N_h^\star}\left[S_h^\star - S_h\right] \geq 0$. Therefore, the disease-free equilibrium *DFE* is globally asymptotically stable.

## Numerical simulation results

In this section, we analyzed the numerical simulation of the proposed model. We used literature values and MATLAB to explore the control reproduction number, the endemic and disease-free equilibrium, and the spread of the Chagas disease. The initial conditions of the state variables are given to be $S_h(0) = 20000$, $I_a(0) = 2000$, $I_c(0) = 4000$, $M(0) = 0$, $S_v(0) = 500000$, and $I_v(0) = 100000$. The baseline parameter values are in Table 2.

### Exploring impact of $\alpha$ on the control reproduction number

First, we examined how varying $\alpha$ affects the reproduction number and the infected population. Essentially, $\alpha$ represents the proportion of infants born to infected mothers that undergo successful treatment.

We generated a contour plot to analyze the control reproduction number ($\mathcal{R}_c$) of the model as a function of the proportion of newborns that undergo successful treatment, $\alpha$, and the progression rate, $p$, as displayed in Fig 2. Based on the contour plot result, the $\mathcal{R}_c$ decreases as more newborns to infected individuals are given the therapy. Thus, higher values of $\alpha$ correspond to lower control reproduction numbers. When $\alpha$ is at its minimum (0), the reproduction number is approximately 3.5. On the other hand, when $\alpha$ is at its maximum (1), the reproduction number decreases to below 1.4. This indicates that the control reproduction number decreases as the proportion of newborns that undergo successful treatment increases, resulting in less disease spread. However, this control measure alone cannot eradicate the disease since the control reproduction is above unity.

Also, we considered the control reproduction number ($\mathcal{R}_c$) as a function of the proportion of newborns that undergo successful treatment, $\alpha$, and the transmission rate, $\beta_{vh}$, as shown in

Table 2. Baseline parameter values of the disease model and their sources.

| Parameter | Value | Source |
|---|---|---|
| $\beta_{vh}$ | 0.000032–0.0000096 per day | [23] |
| $\beta_{hv}$ | 0.000012–0.0000036 per day | [23] |
| $\mu_v$ | 0.005 per day | [30] |
| $\mu_h$ | 0.000042 per day | [30] |
| $\delta_h$ | 0.00013–0.00018 per day | [30] |
| $b_h$ | 70/365 per day | [31] |
| $b_v$ | 183.68–551.04 per day | [32] |
| $k$ | 0.02675 | [30] |
| $p$ | 0.0125 | Assumed |

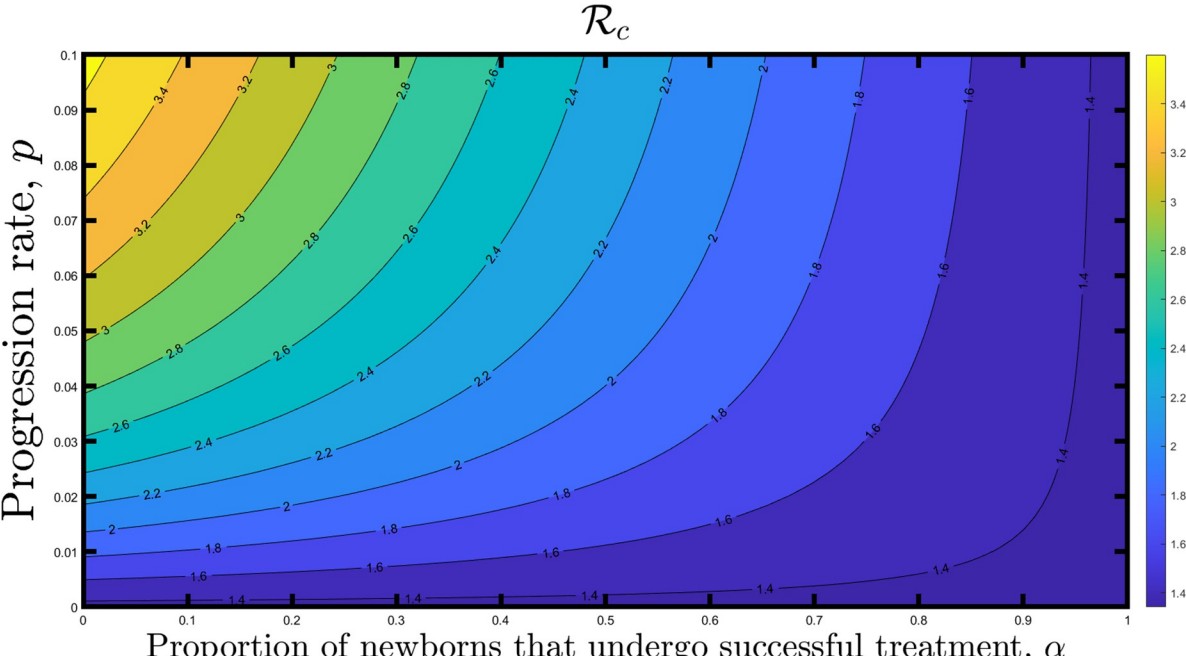

**Fig 2. Contour plot of the control reproduction number ($\mathcal{R}_c$) of the model as a function of the proportion of newborns that undergo successful treatment, $\alpha$, and the progression rate, $p$.**

Fig 3, to examine combinations of the two parameters that are capable of reducing the reproduction number to a desired level. Essentially, effective vector controls lower the $\beta_{vh}$ and reduce the transmission of the infection to susceptible humans. As shown in Fig 3, high values of $\alpha$ and $\beta_{vh}$ cannot reduce the reproduction to below unity. A low $\beta_{vh}$ and a high $\alpha$ are required to control Chagas disease transmission effectively. When $\beta_{vh}$ is set to zero or relatively small, and at least 55% of the newborns successfully receive treatment, the reproduction number drops below unity, emphasizing the importance of vector control and successful newborn therapy measures in reducing the spread and impact of Chagas disease.

### Exploring impact of $\alpha$ on the disease spread

In this section, we used the model to conduct numerical simulations depicting Chagas disease's dynamics. We considered when the system approaches an endemic and disease-free equilibrium, allowing us to observe how the proportion, $\alpha$, of newborns that undergo successful therapy influences the Chagas disease dynamics. We examined how varying $\alpha$ affects the acutely infected population. Again, a low $\alpha$ implies a small proportion of newborns receive successful treatment, and a high $\alpha$ indicates that a significant portion or even all newborns receive successful therapy. The results provided valuable insights for optimizing control strategies for the disease.

Firstly, we considered the impact of $\alpha$ on the infected individuals at an endemic equilibrium. We set $\alpha = 0.5$ (the baseline scenario) and then modified it by increasing and decreasing its value by 25%, 50%, and 75% to examine how varying $\alpha$ values affected the spread, particularly when the system approaches an endemic equilibrium. This led to seven different $\alpha$ values: the baseline $\alpha$ of 0.5, a 25% increase ($\alpha = 0.625$), a 25% decrease ($\alpha = 0.375$), a 50% increase ($\alpha = 0.75$), a 50% decrease ($\alpha = 0.25$), a 75% increase ($\alpha = 0.875$), and a 75% decrease ($\alpha = 0.125$).

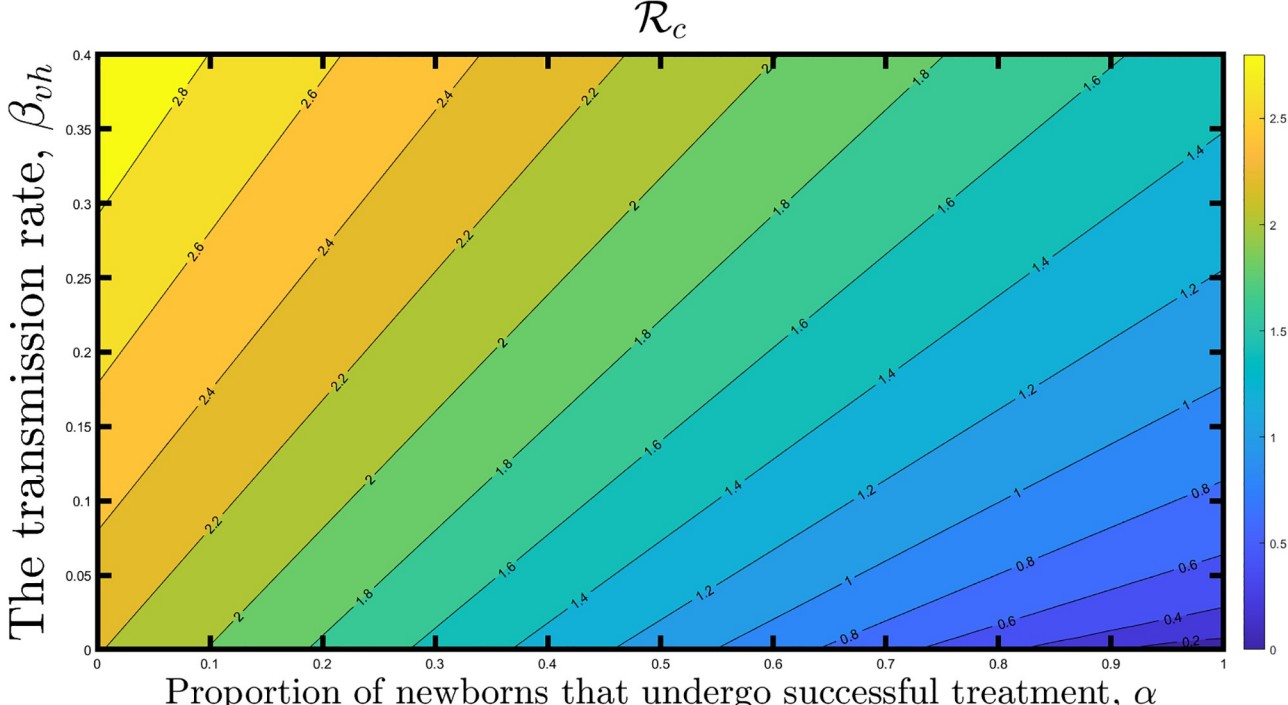

**Fig 3. Contour plot of the control reproduction number ($\mathcal{R}_c$) of the model as a function of the proportion of newborns that undergo successful treatment, $\alpha$, and the transmission rate, $\beta_{vh}$.**

We also looked at the scenarios where $\alpha$ was set to 1 (universal treatment) and 0 (no newborn receives treatment). Fig 4 illustrates the population of acutely infected individuals over the time interval [0, 3000], given these $\alpha$ values. We calculated the area under the curve (AUC), providing a measure of the acutely infected population over time for each $\alpha$ value. This analysis assumes an endemic equilibrium, where the disease remains consistently present in the population over a long period. The calculated percentage changes represent the difference in the infected population compared to the baseline scenario, illustrating the impact of adjusting $\alpha$ values on disease dynamics. The respective AUC values for the different scenarios and percentage change values are presented in Table 3.

Interpreting these results, a 25% increase in $\alpha$ correlated with a 4.79% decrease in the acutely infected population relative to the baseline. Conversely, a 25% decrease in $\alpha$ corresponded to a 4.94% increase in the infected population. Similarly, a 50% increase in $\alpha$ resulted in a 9.43% reduction in the infected population, signifying significant progress in disease management. Meanwhile, a 50% decrease in $\alpha$ led to a 10.03% increase in the infected population. A 75% increase in $\alpha$ correlated with a 13.93% reduction in the infected population, while a 75% decrease in $\alpha$ resulted in a 15.28% increase in infections. When $\alpha$ was set to 1 (100% increase, universal treatment), there was an 18.29% reduction in the acutely infected population, whereas setting $\alpha$ to 0 (100% decrease, no newborn receives treatment) resulted in a 20.69% increase in the acutely infected population. These results underscore how $\alpha$, representing the proportion of newborns that receive treatment, influences the disease spread. Higher $\alpha$ values lead to reduced endemic levels, while lower values indicate higher endemic levels due to a smaller proportion of newborns receiving treatment.

Next, we explored the effect of $\alpha$ at the disease-free equilibrium. In this case, the disease is absent in the population. To accomplish disease-free equilibrium for our simulations, we

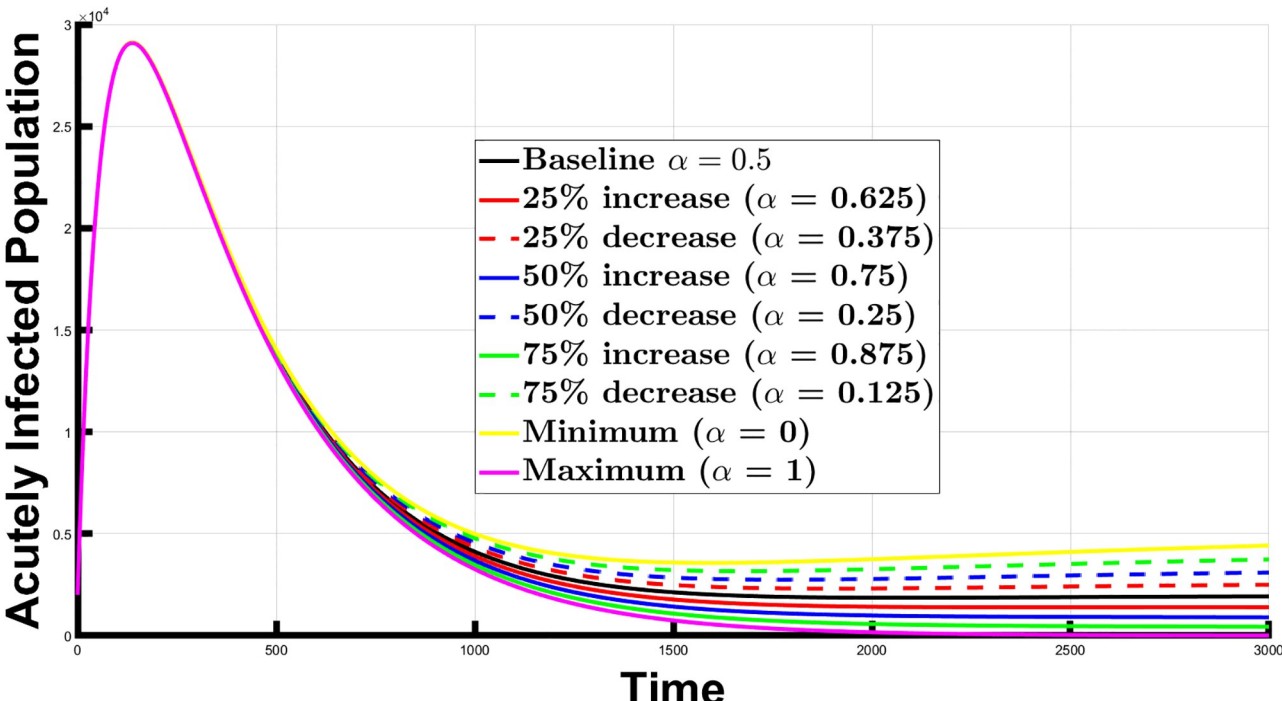

**Fig 4. The acutely infected population at an endemic equilibrium, for different $\alpha$ values.** The figure illustrates the impact of varying $\alpha$ (proportion of newborns receiving treatment) on disease spread, ranging from baseline ($\alpha = 0.5$) to 25%, 50%, and 75% increases and decreases. It also shows scenarios of universal treatment ($\alpha = 1$) and no treatment ($\alpha = 0$).

modified the values of the transmission rates, $\beta_{vh}$ (transmission rate from infected vectors to susceptible humans) and $\beta_{hv}$ (transmission rate from infected humans to susceptible vectors). Vector-human interactions are linked to these parameters, and implementing efficient vector control measures can reduce their values. The values tested for $\alpha$ were 0.5 (baseline), 0 (100% decrease), and 1 (100% increase). Again, an $\alpha$ of 0 means no newborns receive treatment, while an $\alpha$ of 1 means all newborns receive treatment. Fig 5 shows the population of the acutely infected individuals over a given time interval. We calculated the area under the curve for each $\alpha$ value and determined the percent change from the baseline, as shown in Table 4.

The results show a 5.40% increase in the acutely infected population when $\alpha = 0$ compared to the baseline ($\alpha = 0.5$), whereas there is a 4.93% decrease when $\alpha = 1$. This reduction from

**Table 3. Impact of varying $\alpha$ on the acutely infected population at an endemic equilibrium.**

| $\alpha$ value | AUC | Percentage Change |
|---|---|---|
| 0.5 (baseline) | 19,155,646.82 | - |
| 0.625 (25% increase) | 18,238,085.57 | -4.79 |
| 0.375 (25% decrease) | 20,101,943.08 | 4.94 |
| 0.75 (50% increase) | 17,348,644.31 | -9.43 |
| 0.25 (50% decrease) | 21,077,599.61 | 10.03 |
| 0.875 (75% increase) | 16,486,718.12 | -13.93 |
| 0.125 (75% decrease) | 22,083,252.08 | 15.28 |
| 0 (100% decrease) | 23,119,546.64 | 20.69 |
| 1 (100% increase) | 15,651,712.072 | -18.29 |

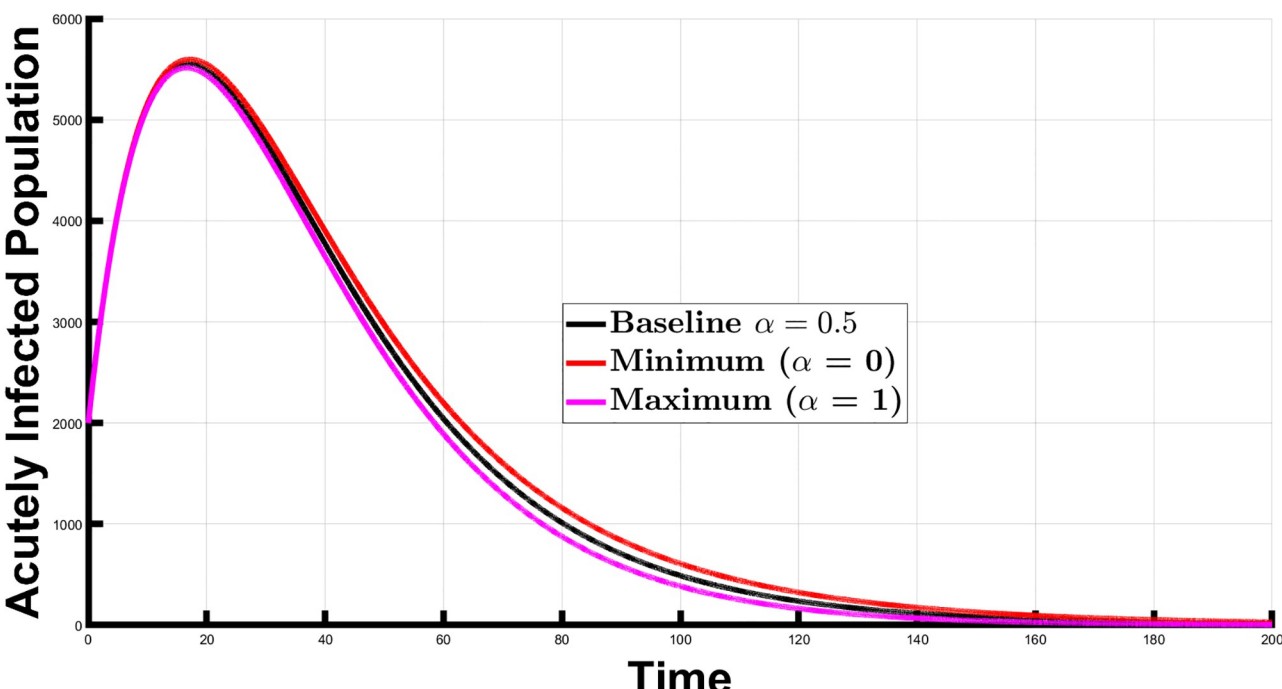

**Fig 5. The number of acutely infected individuals under the disease-free equilibrium scenario of Chagas disease dynamics.** The figure illustrates the impact of varying $\alpha$ values on Chagas disease incidence: baseline ($\alpha = 0.5$), no treatment ($\alpha = 0$), and universal treatment ($\alpha = 1$). In this case, disease-free equilibrium is achieved with a low vector transmission rate.

the baseline suggests that increasing the proportion of newborns receiving treatment effectively reduces the burden of Chagas disease. Thus, a disease-free equilibrium is achievable with both effective vector and newborn therapy control measures.

To explore the impact of $\alpha$ further, we considered the scenario where the system approached a disease-free equilibrium with a specific $\alpha$ value. We compared it to a scenario where $\alpha$ is not introduced, that is, $\alpha = 0$, and the system grows within the comparison period. The results are graphically represented in Fig 6, and values for the AUC and percentage changes from the baseline can be found in Table 5. The graph shows that introducing newborn treatment creates disease-free conditions. Specifically, treating half of the newborns ($\alpha = 0.5$) is sufficient to achieve disease-free equilibrium. But, without any newborn treatment ($\alpha = 0$), the disease continues to grow, which suggests that newborn treatment control is necessary. Additionally, at $\alpha = 0.5$, the system slowly approaches disease-free equilibrium. Increasing $\alpha$ to 1 accelerates this transition, as shown by the smaller AUC (340,787.08) compared to the $\alpha = 0$ (4,522,769.21). This underscores the importance of increasing the number of newborns receiving treatment to control disease spread effectively.

**Table 4. Impact of varying $\alpha$ on the acutely infected population at a disease-free equilibrium.**

| $\alpha$ value | AUC | Percentage Change |
|---|---|---|
| 0.5 (baseline) | 302,553.67 | - |
| 0 (100% decrease) | 318,903.40 | 5.40 |
| 1 (100% increase) | 287,632.49 | -4.93 |

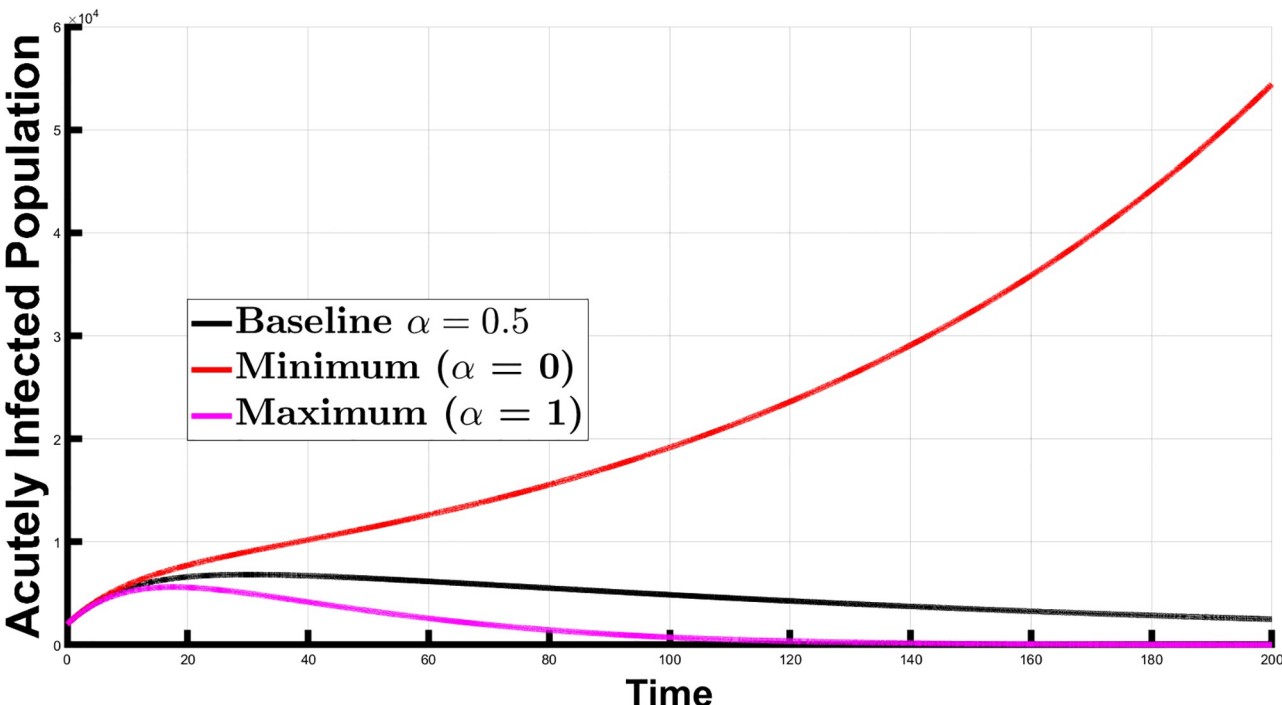

**Fig 6. The number of acutely infected individuals under the disease-free equilibrium with newborn therapy and the disease spread without newborn therapy.**

## Discussion

In this paper, we constructed and analyzed a deterministic Chagas disease model that accounted for the proportion of newborns to infected mothers who undergo treatment therapy. This section discusses the implications of our findings and their significance for strengthening control strategies for Chagas disease. The theoretical and numerical results highlighted that treating a reasonable proportion (or even implementing a universal therapy) of newborns to infected mothers significantly impacts the spread of the disease.

We analyzed the control reproduction number of the model using contour plots. The contour plots showed that the control reproduction number decreases as the proportion of newborns that undergo successful treatment increases, which will result in less disease spread. However, this control measure alone is insufficient to eliminate the Chagas disease. A low vectorial transmission and a high newborn therapy are required to control Chagas disease transmission effectively. It was also observed that if vector transmission can be managed, at least 55% of the babies must receive treatment to eradicate the disease.

In a scenario where endemic equilibrium exists, increasing $\alpha$ from a baseline value 0.5 results in a reduction in the infected population by 4.79%, 9.43%, and 13.93% for 25%, 50%,

**Table 5. Impact of $\alpha$ on the acutely infected population at a disease-free equilibrium, and the disease spread without newborn therapy.**

| $\alpha$ value | AUC | Percentage Change (from Baseline AUC) |
|---|---|---|
| 0.5 (baseline) | 943,004.29 | - |
| 0 (100% decrease) | 4,522,769.21 | 379.61 |
| 1 (100% increase) | 340,787.08 | -63.86 |

and 75% increases in $\alpha$, respectively. Conversely, decreasing $\alpha$ by the same percentages leads to an increase in the infected population by 4.94%, 10.03%, and 15.28%. These findings highlight the importance of treating newborns as it can substantially reduce the burden of Chagas disease. Additionally, the results show the impact of minimizing ($\alpha = 0$) and maximizing ($\alpha = 1$) the proportion of newborns that receive treatment. Minimizing $\alpha$, implying no newborns receive treatment, led to a 20.69% increase in the infected population. On the other hand, maximizing $\alpha$ led to an 18.29% reduction in the acutely infected population.

While our results underscore the significant impact of $\alpha$ in influencing the spread of Chagas disease, it is crucial to recognize that these factors alone are necessary but insufficient for eliminating the disease burden. A more comprehensive approach that includes newborn therapy and vector control strategies, such as insecticide spraying, addressing poor housing conditions, and initiating educational programs to reduce human-vector contact, is crucial to combat Chagas disease effectively. This is important because Chagas disease primarily spreads through the triatomine bugs, which serve as vectors for the *Trypanosoma cruzi* parasite. These vectors play a pivotal role in disease transmission, and their control is essential for reducing vectorial transmission. Our simulations demonstrate the impact of vector control by considering disease-free equilibrium scenarios. In these cases, the values of $\beta_{vh}$ and $\beta_{hv}$ are reduced to smaller values. When no newborn receives treatment ($\alpha = 0$), there was a 5.40% rise in acutely infected individuals, whereas treating all newborns ($\alpha = 1$) resulted in a 4.93% reduction in the acutely infected population. These results underscore how essential vector control measures are to reduce human-vector interactions and their transmission rates.

Based on our results, it is clear that a multifaceted approach, including increasing newborn treatment and implementing vector control measures, is imperative for managing Chagas disease. Treating newborns is very important for controlling and reducing the burden of Chagas disease; however, that does not address vector-borne transmission. Hence, it becomes clear that vector control has to be part of the multifaceted approach to mitigate Chagas disease transmission. Public health interventions should consider these varied disease control methods to develop exhaustive strategies regulating both congenital and vector-to-human transmission routes. We recommend initiatives that raise awareness about Chagas disease and promote early diagnosis and treatment. With these combined efforts, the burden of Chagas disease can be significantly reduced, protecting many people from its harmful effects.

Our study has some limitations that should be acknowledged. This model did not incorporate the impacts of treating Chagas disease-infected individuals, particularly potential mothers. However, including treatment for infected potential mothers may prevent transmission to their progeny. Additionally, our model relied on existing data for parameter values, including the transmission rates. The availability of comprehensive data could help estimate the parameter values and validate the model. Also, Chagas disease transmission is complex; our model is simplified and does not account for other transmission routes and domestic animals that contribute to the disease. Despite these limitations, our study has many strengths. We provide essential insights into the congenital transmission of Chagas disease, highlighting the importance of therapy to newborns of infected mothers. These results are significant as they inform policy decisions for enhancing Chagas disease control.

## Author Contributions

**Conceptualization:** Meriem Boukaabar, Bismark Oduro, Paul Chataa.

**Formal analysis:** Meriem Boukaabar, Bismark Oduro, Paul Chataa.

**Methodology:** Meriem Boukaabar, Bismark Oduro, Paul Chataa.

**Supervision:** Bismark Oduro.

**Writing – original draft:** Meriem Boukaabar.

**Writing – review & editing:** Bismark Oduro.

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
