## [Decision Letter · Decision Letter 0]

12 Apr 2024

PONE-D-24-06545Congenital transmission of Chagas disease: The role of newborn therapy on the disease’s dynamicsPLOS ONE

Dear Dr. Chataa,

Thank you for submitting your manuscript to PLOS ONE. After careful consideration, we feel that it has merit but does not fully meet PLOS ONE’s publication criteria as it currently stands. Therefore, we invite you to submit a revised version of the manuscript that addresses the points raised during the review process.

The paper have been reviewer by two reviewers and both suggest a number of important changes. I agree with their recommendations and hope the authors can address the reviewer's concerns and suggestions in their major revision.

We look forward to receiving your revised manuscript.

Kind regards,

Jan Rychtář

Academic Editor

PLOS ONE

Journal Requirements:

Additional Editor Comments:

The paper have been reviewer by two reviewers and both suggest a number of important changes. I agree with their recommendations and hope the authors can address the reviewer's concerns and suggestions in their major revision.

Reviewers' comments:

Reviewer's Responses to Questions

**Comments to the Author**

1. Is the manuscript technically sound, and do the data support the conclusions?

Reviewer #1: Partly

Reviewer #2: Yes

2. Has the statistical analysis been performed appropriately and rigorously? 

Reviewer #1: N/A

Reviewer #2: I Don't Know

3. Have the authors made all data underlying the findings in their manuscript fully available?

Reviewer #1: Yes

Reviewer #2: Yes

4. Is the manuscript presented in an intelligible fashion and written in standard English?

Reviewer #1: No

Reviewer #2: Yes

5. Review Comments to the Author

Reviewer #1: This is a relatively standard ODE model of a disease. The model is appropriate for Chagas disease and the analysis is sound.

The biggest issue with this paper lies in the result section. The results and the figures are presented in a cumbersome manner. Results on the effect of alpha should be presented in a table and also a figure (with change of alpha on the x axis and the effect on the y axis)

An even more important issue is that the authors consider only DFE and not endemic equilibrium. Their proofs for stability of DFE are standard and can perhaps be streamlined. The formulas for EE are missing completely and should be added.

The validation of the model is missing, i.e., there is no clear reason to believe that the model fits reality. In fact, the model most likely does not fit the reality as it predicts an exponential growth. I would be inclined to believe that in most regions with Chagas, the disease is already endemic and more or less on a stable levels (or fluctuating due to environmental and other changes) but not exponentially growing.

Due to the above issue, the EE should be evaluated, the model validated, perhaps compared to EE and then the sensitivity to alpha reconsidered (i.e. describe how alpha influences the EE, not the growth)

Reviewer #2: Dear authors, this is a study highlighting the importance of adressing the importance of newborn therapy for Chagas Disease control. The methods are sound and the results interesting. There are but a few comments I would like to do.

1. I would recommend abbreviating the introduction.

2. I would add abbreviation definitions to the footnote of figure 1.

3. I would end the introduction section stating the study question.

4. My biggest concern, however, is the fact that treatment of infected humans with Chronic Chagas Disease has not been included in the model. I would think this is very likely to have an important impact on the congenital transmission for two reasons, more infected mothers will be treated and thus would not transmit the disease and because as less infected humans are present less infections of the bugs would occur. I believe this is an important caveat and should be addressed in the discussion.

5. I also miss a section evaluating the limitations and strengths of the conclusions reached.

6. PLOS authors have the option to publish the peer review history of their article (what does this mean?). If published, this will include your full peer review and any attached files.

Reviewer #1: No

Reviewer #2: No

---

## [Author Response · Author response to Decision Letter 0]

20 May 2024

Dear Editor:

The authors would like to thank you for the opportunity to revise our manuscript. We also want to thank both reviewers and the Handling Editor for thoughtful comments and suggestions on the previous version of this manuscript. We have thoroughly revised it and addressed their comments. Below, please find our response to the reviewers' comments. 

REVIEWER 1:

We appreciate the reviewer's feedback and do address the concerns below.

 This is a relatively standard ODE model of a disease. The model is appropriate for Chagas disease and the analysis is sound.

The biggest issue with this paper lies in the result section. The results and the figures are presented in a cumbersome manner. Results on the effect of alpha should be presented in a table and also a figure (with change of alpha on the x axis and the effect on the y axis)

We thank the reviewer for this comment. We have summarized the impact of the alpha on the infected individual's compartment of the model in Table 3. Furthermore, we have added two new plots: Figure 2(a) (contour plot), and Figure 2(b) (I^*-vs alpha) to further demonstrate the effect of alpha on the model dynamics. 

An even more important issue is that the authors consider only DFE and not endemic equilibrium. Their proofs for stability of DFE are standard and can perhaps be streamlined. The formulas for EE are missing completely and should be added.

We thank the reviewer for this comment. We have streamlined the local stability proof of DFE.

The validation of the model is missing, i.e., there is no clear reason to believe that the model fits reality. In fact, the model most likely does not fit the reality as it predicts an exponential growth. I would be inclined to believe that in most regions with Chagas, the disease is already endemic and more or less on a stable levels (or fluctuating due to environmental and other changes) but not exponentially growing.

Due to the above issue, the EE should be evaluated, the model validated, perhaps compared to EE and then the sensitivity to alpha reconsidered (i.e. describe how alpha influences the EE, not the growth)

We thank the reviewer for this comment. This is an ODE model that grows, decays, and reaches an equilibrium under certain conditions. We have evaluated and derived conditions for which the EE exists. We have also analyzed the impact of the alpha on the EE both numerically and theoretically. Figure 5(b) (I^*-vs alpha) demonstrates the effect of alpha on the EE and these results collaborate. 

REVIEWER 2:

We thank the reviewer for carefully reading the manuscript and for valuable suggestions. We have incorporated the suggestions in the revised version.

Dear authors, this is a study highlighting the importance of adressing the importance of newborn therapy for Chagas Disease control. The methods are sound and the results interesting. There are but a few comments I would like to do.

1. I would recommend abbreviating the introduction.

We thank the reviewer for this comment. We agree with the reviewer that the introduction was long and have reduced it by one paragraph.

2. I would add abbreviation definitions to the footnote of figure 1.

We thank the reviewer for this comment. However, the comment is not clear to us. Figure I is the schematic diagram of the model. The model parameters are defined in Table 1 and the variables are defined in the first paragraph of the materials and methods section. 

3. I would end the introduction section stating the study question.

We thank the reviewer for this comment. We have rephrased the last paragraph of the introduction and added the research question.

4. My biggest concern, however, is the fact that treatment of infected humans with Chronic Chagas Disease has not been included in the model. I would think this is very likely to have an important impact on the congenital transmission for two reasons, more infected mothers will be treated and thus would not transmit the disease and because as less infected humans are present less infections of the bugs would occur. I believe this is an important caveat and should be addressed in the discussion.

We thank the reviewer for this comment. Several studies have confirmed the effectiveness of acutely infected individuals and are less effective for chronic Chagas disease patients. Nevertheless, we have addressed this comment as a limitation. 

5. I also miss a section evaluating the limitations and strengths of the conclusions reached

We thank the reviewer for this comment. We have added a paragraph addressing limitations and strengths.

---

## [Decision Letter · Decision Letter 1]

29 May 2024

PONE-D-24-06545R1Congenital transmission of Chagas disease: The role of newborn therapy on the disease’s dynamicsPLOS ONE

Dear Dr. Chataa,

Thank you for submitting your manuscript to PLOS ONE. After careful consideration, we feel that it has merit but does not fully meet PLOS ONE’s publication criteria as it currently stands. Therefore, we invite you to submit a revised version of the manuscript that addresses the points raised during the review process.

The reviewer still finds several major issues that need to be fixed. All of the raised points are important and need to be addresses, starting with point #3 that identifies a problem with the model setup, point #1 that asks for more formulas so that readers can follow the calculations for the endemic equilibrium (plus a fact that R0 should most likely pop up in the formulas), and finally point #2 about simulating the appropriate scenarios instead of starting near disease-free equilibrium.

We look forward to receiving your revised manuscript.

Kind regards,

Jan Rychtář

Academic Editor

PLOS ONE

Additional Editor Comments:

The reviewer still finds several major issues that need to be fixed. All of the raised points are important and need to be addresses, starting with point #3 that identifies a problem with the model setup, point #1 that asks for more formulas so that readers can follow the calculations for the endemic equilibrium (plus a fact that R0 should most likely pop up in the formulas), and finally point #2 about simulating the appropriate scenarios instead of starting near disease-free equilibrium.

Reviewers' comments:

Reviewer's Responses to Questions

**Comments to the Author**

1. If the authors have adequately addressed your comments raised in a previous round of review and you feel that this manuscript is now acceptable for publication, you may indicate that here to bypass the “Comments to the Author” section, enter your conflict of interest statement in the “Confidential to Editor” section, and submit your "Accept" recommendation.

Reviewer #1: (No Response)

2. Is the manuscript technically sound, and do the data support the conclusions?

Reviewer #1: Partly

3. Has the statistical analysis been performed appropriately and rigorously? 

Reviewer #1: N/A

4. Have the authors made all data underlying the findings in their manuscript fully available?

Reviewer #1: No

5. Is the manuscript presented in an intelligible fashion and written in standard English?

Reviewer #1: (No Response)

6. Review Comments to the Author

Reviewer #1: Thank you for revising the manuscript. There are still several issues that should be addressed

1) The authors provide formulas for the endemic equilibrium but not the derivation. Some calculations should be provided that would indicate that the formulas are correct. It is common that the formulas for EE contains basic reproduction number and that one needs the reproduction number to be greater than 1 for the formulas to be biologically reasonable. The provided formulas are sort of along the lines, but it is really not clear that they are correct. So, my suggestion is to a) include the calculations and b) look for the reproduction number in the formulas.

2) The figures still show exponential growth (i.e. what happens when one varies alpha in the DFE). As mentioned earlier, the diseases is endemic. So it would make more sense to investigate what happens when one varies alpha in EE

in light of the point 3 below, if the authors indeed change the model, this point be addressed by varying r rather than alpha (assuming omega is more or less given by the nature of the treatment).

3) upon closer reading of the model description, there appears to be a very important and significant inconsistency in how authors treat alpha and p and r. They introduce alpha as omega times r where omega is treatment efficacy and r is a treatment rate. That means that alpha is a rate. The term alpha times p times M for the transmission from M to Sh does not make much sense (one cannot multiply rate with a rate to get a rate). Similarly, the term (1-alpha) in the flow from M to Ia does not make much sense either (on its own and definitely not when multiplied again by a rate p). It seems that much more natural rates from M to Sh would be omega times r and from M to Ia to be (1-omega) times r. There is also probably another process (such as aging) during which the untreated newborns from M become Ia. This is likely why authors used p in the model (and the overall rate from M to Ia would then be p+(1-omega)r).

So, my suggestions is for the authors to revise the model appropriately (or if they do not agree with the above interpretation, explain the model properly)

7. PLOS authors have the option to publish the peer review history of their article (what does this mean?). If published, this will include your full peer review and any attached files.

Reviewer #1: No

---

## [Author Response · Author response to Decision Letter 1]

5 Jul 2024

Dear Editor:

The authors would like to thank you again for the opportunity to revise our manuscript. We also want to thank the reviewer for thoughtful comments and suggestions on the previous version of this manuscript. We have thoroughly revised it and addressed their comments. Below (in blue), please find our response to the reviewers' comments. 

REVIEWER 1:

Reviewer #1: Thank you for revising the manuscript. There are still several issues that should be addressed

1) The authors provide formulas for the endemic equilibrium but not the derivation. Some calculations should be provided that would indicate that the formulas are correct. It is common that the formulas for EE contains basic reproduction number and that one needs the reproduction number to be greater than 1 for the formulas to be biologically reasonable. The provided formulas are sort of along the lines, but it is really not clear that they are correct. So, my suggestion is to a) include the calculations and b) look for the reproduction number in the formulas.

We thank the reviewer for this comment and suggestion. We have added some details and streamlined the derivation of the EE. We also agree with the reviewer that the EE of some models can be expressed in terms of the reproduction number; unfortunately, our EE cannot be written in terms of our R_c/R_L. 

2) The figures still show exponential growth (i.e. what happens when one varies alpha in the DFE). As mentioned earlier, the diseases is endemic. So it would make more sense to investigate what happens when one varies alpha in EE.

We thank the reviewer for the comment. We have provided simulations showing the effects of alpha on the system at EE (Figure 4) and DFE (Figure 5).

in light of the point 3 below, if the authors indeed change the model, this point be addressed by varying r rather than alpha (assuming omega is more or less given by the nature of the treatment).

3) upon closer reading of the model description, there appears to be a very important and significant inconsistency in how authors treat alpha and p and r. They introduce alpha as omega times r where omega is treatment efficacy and r is a treatment rate. That means that alpha is a rate. The term alpha times p times M for the transmission from M to Sh does not make much sense (one cannot multiply rate with a rate to get a rate). Similarly, the term (1-alpha) in the flow from M to Ia does not make much sense either (on its own and definitely not when multiplied again by a rate p). It seems that much more natural rates from M to Sh would be omega times r and from M to Ia to be (1-omega) times r. There is also probably another process (such as aging) during which the untreated newborns from M become Ia. This is likely why authors used p in the model (and the overall rate from M to Ia would then be p+(1-omega)r).

So, my suggestions is for the authors to revise the model appropriately (or if they do not agree with the above interpretation, explain the model properly)

We thank the reviewer for this comment. This is important feedback, and we agree that the parameter alpha was not well-defined. We have redefined and explained alpha. We also thank the reviewer for the suggested model. It is a great idea, and we thoroughly explored it. However, we observed some concerns and caveats in this version of the model. For example, while I_a decreases with respect to omega (efficacy), it increases with respect to the treatment rate, r, with a fixed omega.

---

## [Decision Letter · Decision Letter 2]

16 Jul 2024

Congenital transmission of Chagas disease: The role of newborn therapy on the disease’s dynamics

PONE-D-24-06545R2

Dear Dr. Chataa,

We’re pleased to inform you that your manuscript has been judged scientifically suitable for publication and will be formally accepted for publication once it meets all outstanding technical requirements.

Kind regards,

Jan Rychtář

Academic Editor

PLOS ONE

Additional Editor Comments (optional):

All comments have been addressed, thank you!

Reviewers' comments:

Reviewer's Responses to Questions

**Comments to the Author**

1. If the authors have adequately addressed your comments raised in a previous round of review and you feel that this manuscript is now acceptable for publication, you may indicate that here to bypass the “Comments to the Author” section, enter your conflict of interest statement in the “Confidential to Editor” section, and submit your "Accept" recommendation.

Reviewer #1: All comments have been addressed

2. Is the manuscript technically sound, and do the data support the conclusions?

Reviewer #1: Yes

3. Has the statistical analysis been performed appropriately and rigorously? 

Reviewer #1: N/A

4. Have the authors made all data underlying the findings in their manuscript fully available?

Reviewer #1: Yes

5. Is the manuscript presented in an intelligible fashion and written in standard English?

Reviewer #1: Yes

6. Review Comments to the Author

Reviewer #1: The authors addressed all the comments.

7. PLOS authors have the option to publish the peer review history of their article (what does this mean?). If published, this will include your full peer review and any attached files.

Reviewer #1: No

---

## [Editor Report · Acceptance letter]

23 Jul 2024

PONE-D-24-06545R2 

PLOS ONE

Dear Dr. Chataa, 

I'm pleased to inform you that your manuscript has been deemed suitable for publication in PLOS ONE. Congratulations! Your manuscript is now being handed over to our production team.

Kind regards, 

on behalf of

Dr. Jan Rychtář 

Academic Editor

PLOS ONE